# OpenReview forum: "When Thinking Backfires: Mechanistic Insights into Reason-induced Misalignment"
_ICLR.cc/2026/Conference — ICLR 2026 Poster_

### Official Review · Reviewer_VT3w · 2025-10-31

**Soundness:** 3
**Presentation:** 3
**Contribution:** 3
**Rating:** 6
**Confidence:** 4

**Summary:**

This paper investigates the phenomenon of Reasoning-Induced Misalignment (RIM) in large language models (LLMs), where enhancing models' reasoning abilities, particularly through chain-of-thought (CoT) prompting or fine-tuning on reasoning tasks, paradoxically results in increased susceptibility to malicious prompts. The authors offer a detailed mechanistic study, identifying specific attention heads and safety-critical neurons whose representational changes and entanglement underlie the observed trade-off between reasoning and safety. A novel metric, Reciprocal Activation Shift (RAS), is proposed to quantify and predict catastrophic forgetting at the neural level, correlating representational entanglement with misalignment outcomes.

**Strengths:**

- The paper studies RIM across a diverse set of open-source LLMs (dense and MoE), employing multiple math-reasoning datasets and a rigorous safety benchmark (HEx-PHI). This breadth lends robust credibility to the main claims, as captured in Table 1 and 2, and in the various controlled experiments across inference and training.
- The work employs unsupervised probing, attention-head ablations, and neuron-level causal interventions to connect behavioral safety failures directly to internal model components. For example, Figure 3 and 4 present clear layer-wise probe score visualizations, supporting the hypothesis that co-attention on CoT tokens and refusal dynamics are intricately linked.
- The introduction of the RAS metric and its application to both safety-critical and random neurons is a notable methodological addition. Table in the RAS section and Figure 9 demonstrate that RAS outperforms KL-divergence and other baselines in predicting catastrophic forgetting and safety–reasoning trade-offs. The various ablation studies (e.g., Figure 8) solidify the claim that specific neural circuits are disproportionately affected by reasoning-focused fine-tuning.

**Weaknesses:**

- All core misalignment results hinge on HEx-PHI, a single benchmark for safety evaluation. This is explicitly acknowledged in the limitations, but it introduces significant risk of dataset overfitting or unmeasured generalization failure. Evaluating on at least one additional, established safety benchmark would strengthen claims about the universality of RIM.
- While the focus on math tasks is motivated, the implication is that RIM might extend to other reasoning domains (e.g., commonsense, coding, multi-step logic), but this is not empirically tested. Even a brief pilot evaluation in another domain would aid in generalizing the trade-off claims.
- The methodology for constructing "effort-minimizing" reasoning patterns and their control groups is defined through LLM editing prompts, but further quantification of linguistic/semantic divergence between control and target CoTs would clarify causal attribution. At present, it is possible that length or unrelated stylistic features still confound the effect (although authors attempt to control for length).

**Questions:**

- Can the authors provide empirical results on at least one additional safety/misalignment benchmark to validate that RIM and associated mechanisms generalize? Even a small-scale or qualitative demonstration would increase confidence.
- What methodological rationale justifies the harmonic mean in the RAS metric? Would alternative aggregation (arithmetic, geometric, subtraction, or ratio) yield similar or different outcomes? Ablation on this point would clarify the reliability of entanglement quantification.
- How well do the proposed mechanisms and entanglement metrics transfer to domains beyond math reasoning—such as logical reasoning, coding, or commonsense? Are similar safety-effort trade-offs and attention/neuron patterns observed?

---

> ### Author Response · Authors · 2025-11-20
> **Response Part 1**
>
> Thank you for the valuable comments and suggestions. We have revised and included some results and we hope the revision could address your concerns.
>
> ### Q1: new safety benchmarks
> We added two new safety benchmarks: HarmBench[1] and AgentHarm[2]. The new results for off-the-shelf and fine-tuned models are as follows:
> > __Results for off-the-shelf models in Table 1 and Table 2.__
> > - For HarmBench, all Qwen3 models except Qwen3-30B-A3B have worsened alignment rate when using think-mode, with Qwen3-32B in particular where the misalignment rate under think-mode is 6.68x that of no-think mode.
> > - For AgentHarm, all Qwen3 models suffer more severe misalignment when think-mode is enabled.
> >
> > The new results demonstrate that RIM is prevalent across multiple established alignment benchmarks. We have added these results in __Table A2__ of the revised manuscript.
> >
> > &nbsp;
> >
> > __[Table 1: Misalignment rates on HarmBench with different think modes during model inference.]__
> | Think-Mode | Qwen3-4B | Qwen3-8B | Qwen3-32B | Qwen3-30B-A3B |
> | -------- | -------- | -------- | -------- | -------- |
> | On | 22.61% | 19.00%  | 30.21% | 15.30% |
> | OFF | 13.50% | 14.50% | 4.52% | 10.50% |
> >
> > &nbsp;
> >
> > __[Table 2: Misalignment rates on AgentHarm with different think modes during model inference.]__
> | Think-Mode | Qwen3-4B | Qwen3-8B | Qwen3-32B | Qwen3-30B-A3B |
> | -- | -- | -- | -- | -- |
> | On | 30.77% | 26.92% | 28.85% | 21.15% |
> | OFF | 25.00% | 23.08% | 13.46% | 17.31% |
>
> > __Results for fine-tuned models in Table 3 and Table 4.__
> >
> > Evaluation results on both HarmBench and AgentBench show that fine-tuning on math, logic, and commonsense reasoning tasks generally leads to worsened alignment. These results, combined with the existing results from HEx-PHI, validates our discovery of Reasoning-Induced Misalignment and confirms that RIM extends to a wide range of safety benchmarks. The results have been added to __Table A9 and A10__ in the revised manuscript.
> >
> > &nbsp;
> >
> > __[Table 3: Changes of misalignment rate results on HarmBench after fine-tuning on reasoning tasks.]__
> | Model | MATH401 | MATH500 | GSM8k | PrOntoQA | StrategyQA |
> | -- | -- | -- | -- | -- | -- |
> | Qwen3-4B | 4.68% | 1.00% | 2.91% | 15.37% | 8.50% |
> | Phi3.5-Mini | 7.50% | 1.50% | 10.61% | 22.16% | 11.50% |
> | OLMo2-1B |-5.56% |2.94% |6.03% |0.50% | -2.56% |
> | Mistral-7B |1.46% |0.98% |2.98% |3.01% |3.49% |
> | __Average__ | __2.02%__ | __1.61%__ | __5.63%__ | __10.27%__ | __5.23%__ |
> >
> > &nbsp;
> >
> > __[Table 4: Changes of misalignment rate results on AgentHarm after fine-tuning on reasoning tasks.]__
> | Model | MATH401 | MATH500 | GSM8k | PrOntoQA | StrategyQA |
> | -- | -- | -- | -- | -- | -- |
> | Qwen3-4B |1.92% |8.33% |0.00% |11.54% |3.85% |
> | Phi3.5-Mini |-1.92% |0.00% |7.69% |27.87% |7.69% |
> | OLMo2-1B |0.00% |-3.85% |0.00% |-5.77% |0.00% |
> | Mistral-7B |-5.54% |0.00% |7.84% |9.80% |7.84% |
> | __Average__ |  __-1.39%__ | __1.12%__ | __3.88%__ | __10.86%__ | __4.85%__ |
>
>
> ### Q2: new reasoning tasks
> > We have added a logic reasoning task, namely PrOntoQA [3], and a commonsense reasoning task, namely StrategyQA [4] to further enhance generalization of the RIM observation as well as the RAS metric.
> >
> > &nbsp;
> >
> >>  #### __Q2.1: Fine-tuning with new reasoning tasks also exacerbate misalignment__
> >
> > Misalignment rate after fine-tuning on the new reasoning tasks are shown in Table 5.
> >
> > __[Table 5: Changes in misalignment rates after FT on four models, evaluated on HEx-PHI.]__
> | | Logic Reasoning | Commonsense Reasoning |
> | -- | -- | -- |
> | Qwen3-4B | +14.05% | +2.28% |
> | Phi3.5-Mini | +7.17% | +5.33% |
> | Mistral-7B | +7.13% | +8.90% |
> | OLMo2-1B | -5.10% | -4.06% |
> | __Average__ | __+5.81%__ | __+3.11%__ |
> >
> > From Table 5 (updated as __Table A8__ in the revised manuscript), we observe that fine-tuning on both logic and commonsense reasoning datasets lead to an overall worse misalignment situation across the four models.
> >
> > &nbsp;
> >
> >> #### __Q2.2: Attention Head Pattern__
> >
> > Our current attention pattern analysis is on a safety-related benchmark with think/no-think mode, which is independent of the specific reasoning task. Therefore, the newly added reasoning tasks will not update the identification of attention heads.
> >
> > &nbsp;
> >
> >> #### __Q2.3: Entanglement Metric__
> >
> > We calculate the RAS for fine-tuned models on the logic reasoning dataset [3]. The correlation between RAS and delta misalignment rate are shown in Table 6. It is demonstrated that the RAS is a better proxy across multiple models compared with KL. We added the new results in __Table A12__ of the manuscript.
> >
> > __[Table 6: Correlation between proxy metrics and $\Delta$ misalignment rate.]__
> | Metric | Olmo-1B | Qwen3-4B | Phi3-4B | Mistral-7B | __Average__
> | ------ | ------ | ------ | ------ | ------ | ------ |
> | RAS | 0.643 | 0.718 | 0.337 | 0.617 | __0.579__ |
> | KL-Divergence | -0.027 | -0.180 | 0.258 | 0.459 | __0.127__ |

---

> ### Author Response · Authors · 2025-11-20
> **Response Part 2**
>
> ### Q3: Confounders in “Effort-minimizing" reasoning patterns
>
> > To study the effects of potential confounders, we stratify the target and controlled CoTs by confounder values, and examine misalignment rates after training models on each group. Specifically, We conduct an additional ablation study where we redefine both control and target fine-tuning sets using subsets stratified by specific linguistic features:
> > - (1) average parse-tree depth, capturing sentence-level syntactic complexity;
> > - (2) formal-marker density, measuring the use of discourse and functional markers indicative of more formal structure;
> > - (3) long-word ratio
> > - (4) word count
> >
> > From the results in Table 3 and Table 4, we have observed that:
> >
> > - Within group: for both target and controlled CoTs, groups stratified by the same confounder show minimal misalignment rate differences (4-7%), indicating confounders do not systematically affect outcomes within each CoT type.
> >
> > - Between-group comparison: Target CoTs consistently produce significantly higher misalignment rates than controlled CoTs across all confounder strata (e.g., 60% vs 40% for avg-parse-depth groups).
> >
> >The results imply that the tested confounders do not systematically drive misalignment rates. The consistent gap between target and controlled CoTs across all strata provides evidence that effort-minimizing patterns causally lead to greater misalignment.
> >
> > &nbsp;
> >
> > __[Table 3: Misalignment rate of models after training on different groups of target and control CoTs.]__
> | Subset                               | Target CoTs |  Control CoTs |
> | ------------------------------------ | ------ | ------ |
> | avg-parse-depth (small)       | 68.46% | 42.81% |
> | avg-parse-depth (large)       | 64.88% | 38.80% |
> | formal-marker-density (small) | 70.71% | 42.62% |
> | formal-marker-density (large) | 63.85% |40.00% |
> | long-word-ratio (small)       | 62.75% | 41.67% |
> | long-word-ratio (large)       | 69.02% | 39.60% |
> | word-count (small)            | 66.56% | 40.13% |
> | word-count (large)            | 70.47% | 41.81% |
> >
> > We have added this analysis and further discussion in the revised manuscript (__Appendix E.4__).
>
> ### Q4: Other alternative aggregation for RAS
> > Thanks for your great insight!
> >
> > We already have the Arithmetic Mean (0.55) and Geometric Mean (0.59) results in Table 3, ranking as the second and third best metrics after RAS (0.65, harmonic mean), and substantially outperforming the two baselines: RAS-random neuron (0.06) and KL divergence (0.23).
> >
> > __We add the subtraction and ratio in Table 3__, whose correlations with $\Delta$ Misalignment Rate are:
> > - Ratio (safe/math): -0.50
> > - Ratio (math/safe): 0.48
> > - Subtraction (safe-math): -0.52
> > - Subtraction  (math-safe): 0.52
> >
> > All "difference-based" targeted representation metrics (subtraction, ratio, arithmetic/geometric means) consistently outperform both distribution-level (KL) and random baseline metrics. This demonstrates that targeted neuron-level measurements provide more reliable prediction of misalignment than global distributional changes, with RAS achieving the strongest correlation through its specific formulation.
>
> Please let us know if you have further concerns -- thank you!
>
> References:
>
> [1] HarmBench: A Standardized Evaluation Framework for Automated Red Teaming and Robust Refusal.
>
> [2] AgentHarm: A Benchmark for Measuring Harmfulness of LLM Agents.
>
> [3] Language Models Are Greedy Reasoners: A Systematic Formal Analysis of Chain-of-Thought.
>
> [4] Did Aristotle Use a Laptop? A Question Answering Benchmark with Implicit Reasoning Strategies.

---

> > ### Comment · Reviewer_VT3w · 2025-11-25
> > **Thanks for the detailed response.**
> >
> > Dear Authors,
> >
> > Thanks for the detailed response. My main concerns (other safety benchmarks and other reasoning types) have been properly addressed with the new experiment results provided by the authors.
> >
> > I maintain my initial evaluation of this paper and lean to recommend its acceptance.
> >
> > Reviewer VT3w

---

> > > ### Author Response · Authors · 2025-11-25
> > >
> > > Thank you for your thoughtful review and acknowledgment of our additional responses and experiments. We greatly appreciate your support and recommendation.

---

### Official Review · Reviewer_XKN8 · 2025-11-01

**Soundness:** 3
**Presentation:** 3
**Contribution:** 3
**Rating:** 6
**Confidence:** 3

**Summary:**

The paper studies reasoning-induced misalignment (RIM) in both inference-time-induced reasoning and reasoning finetuning.
- For inference-time reasoning, RIM is amplified by inference-time CoT, especially with effort-minimizing reasoning patterns.
- For finetuning, normal CoT data reduces misalignment rate, while effort-minimizing reasoning patterns increases misalignment.

The paper then studies the mechanism at inference-time and during finetuning separately.
- For inference time behaviors, the paper identifies several attention heads across layers that exhibits refusal behavior. Intervening on these heads reduces refusal rates compared to ablation on random heads.
- For finetuning, the paper identifies "safety-critical neurons" (i.e. the ones that show the most difference in activation on refused vs accepted requests) which increase the misalignment rate strongly when zeroed out.

The paper finds that safety and math abilities may be entangled: for finetuning, removing safety-critical neurons also hurts math performance.

To study the extent of catastrophic forgetting, the paper proposes a "reciprocal activation shift" (RAS) score, which computes the Harmonic mean of the amount of activation change in safety and math tasks.
- Empirically, RAS increases after finetuning, which is expected given its definition.
- RAS is also shown to correlates strongly with misalignment rate.

**Strengths:**

- The paper studies the timely question of reasoning-induced misalignment (RIM) and discovers an interesting safety-reasoning trade-off.
- The investigation moves beyond viewing RIM as a type of catastrophic forgetting, and aims to provide mechanistic explanation. It finds "interference", where the same set of neurons can impact both safety and reasoning.
- The paper provides careful empirical investigation, with thorough experimental evidence and discussions on alternative hypotheses.

**Weaknesses:**

I don't have major complaint about the paper.

One comment is that please consider removing unclear languages such as "cognitively demanding situations" or "cognitive effort" -- what does "cognition" mean for LLMs?

**Questions:**

- Can we predict the extent of RIM before training on a task? For example, by measuring the gradient norms on the data?
- Fig 4: how many inputs are considered?
- Fig 3 & 5: why studying refusal rate at particular tokens, rather than averaged over all tokens? Why do we focus on tokens between think tags? Should we ignore the behaviors at other positions?
- Sec 4.1: are activation values always non-negative (because of ReLU)?
- Line 424: should the paragraph title be "RAS predicts misalignment rate", since RAS is by definition measuring forgetting?
- Table 3: do you have explanations on why the correlation measured for Phi-3.5 is much lower overall?

---

> ### Author Response · Authors · 2025-11-20
>
> Thank you for the insightful comments and suggestions.
>
> ### Q1: Unclear term "cognition"
> > We agree that these terms can be misleading for LLMs, and we have replaced this with ___“reasoning efforts”___ in the revised manuscript.
>
> ### Q2: Prediction RIM before training by measuring gradient norm
> > Thank you for this insightful suggestion. Gradient norm is indeed a meaningful influence measure for input data, with smaller norms often associated with better generalization [1]. In principle, we could perform a forward pass on reasoning data and infer misalignment risk—larger gradient norms would indicate greater parameter changes and potentially higher misalignment rates. The possible challenge is to isolate parameter changes that specifically affect safety-reasoning alignment, rather than general task performance.
> >
> > Complementarily, RAS uses changes in activation values across safety and reasoning tasks over safety-critical neurons to predict RIM, which constrains the focus on misalignment-related behavior. We believe that gradient-norm is a very promising direction worthy of further exploration.
>
> ### Q3: why studying refusal rate at particular tokens?
> > In Section 3, we compare think and no-think modes that differ only in their prompts (Figure A1). Our goal is to identify tokens that are critical for refusal behavior.
> >
> > __Firstly__, the primary structural difference between think/no-think prompts is the empty content between think tags in no-think mode.
> >
> > __Secondly__, we observed specific attention heads that preferentially attend to this region in Figure 4. Probe analysis in Figure4 shows that tokens in this region demonstrate the strongest capability for generating refusal responses, indicating they are mechanistically important for safety behavior.
> >
> > Rather than spreading our analysis across all positions, we focus on the region where the safety behavior is most prominent, allowing us to isolate the specific contribution of the no-think structure to refusal behavior.
>
> ### Q4: Figure4 how many inputs?
> > - For Figure4(a), we show the attention pattern on one sample. Due to page limit, additional examples are provided in Appendix Figure A4.
> >
> > - For Figure4(b), we show the distribution of such refusal heads across 180 samples.  Details can be found in the __updated Appendix C.3.__ We have 180 training samples and 120 evaluation samples from HEx-PHI, i,e., harmful inputs with either fulfill or reject responses. We identify the refusal attention heads (shown in Figure4(b)) from the training set with 180 samples.
> >
> > - To validate that these identified heads systematically govern refusal behavior rather than cherry-picky, we intervene these attention heads identified in Figure4(b) and observe  a substantial degradation in refusal responses (120 samples in the evaluation set) shown in Figure 5.
>
> ### Q5: Activation values are always positive in section 4.1
> > - Thank you for this important question. Activation values can be negative, as the models we study use smoothed activations such as GELU or SiLU.
> >
> > - We apologize for the error in Section 4.1—the absolute value symbol was missing from __the first equation__ though we correctly applied absolute values in our RAS implementation. Since we aim to calculate relative changes (which should be positive values), the absolute value is necessary. We have corrected this in the revised manuscript.
>
> ### Q6:  Paragraph title in Section 4.2
> > Thank you for this suggestion! We have updated the title in the revised manuscript.
> >
> > &nbsp;
> >
> > RAS is a representation-level metric introduced as a proxy for catastrophic forgetting (Section 4.1), which we then empirically validate (Section 4.2). Catastrophic forgetting is defined as performance degradation on the original task [2]; in our safety alignment context, this corresponds to misalignment rate changes/safety degradation on safety benchmarks.
>
> ### Q7: Explanation about correlation for Phi-3.5
> > Our correlation between RAS and catastrophic forgetting is calculated across multiple training checkpoints.
> > - In early training phases, we observe that Phi-3.5 exhibits unusual robustness to safety degradation. Specifically, while other models exceed 90% relative misalignment rate changes, Phi-3.5 reaches only 53% despite experiencing significant representational changes.
> >
> > - This can be attributed to Phi-3.5's superior resistance to safety-related catastrophic forgetting. As stated in the technical report, the Phi-3 model family (including 3.5) underwent a rigorous enhancement process, incorporating both SFT and DPO to ensure precise instruction adherence and robust safety measures.
>
> Please let us know if you have further concerns. Thank you!
>
> References:
>
> [1] Penalizing Gradient Norm for Efficiently Improving Generalization in Deep Learning
>
> [2] Rl’s Razor: Why Online Reinforcement Learning Forgets Less.

---

> > ### Comment · Reviewer_XKN8 · 2025-11-26
> >
> > Thank you for the detailed response! I remain my original score.
> >
> > One clarification: Reference [1] included in the rebuttal is not about the gradient norm, but the weight norm.

---

> > > ### Author Response · Authors · 2025-11-26
> > >
> > > Thanks for your reply and clarification!
> > >
> > > We have updated the reference [1] as "Penalizing Gradient Norm for Efficiently Improving Generalization in Deep Learning"

---

### Official Review · Reviewer_mDjh · 2025-11-01

**Soundness:** 3
**Presentation:** 3
**Contribution:** 3
**Rating:** 8
**Confidence:** 4

**Summary:**

This paper identified and mechanistically analyzed a concerning phenomenon: Reasoning-Induced Misalignment (RIM). It discovers that safety alignment could decrease when reasoning capabilities strengthen. Based on experiments in in-context CoT prompting and finetuning on reasoning tasks, the paper discovers that effort-minimizing reasoning patterns could lead to the RIM phenomenon. Then, the paper tries to understand the RIM phenomenon from the neuron level. For in-context CoT inference, it discovers that CoT changes the behavior of some attention heads, which induces safety misalignment. For fine-tuning, the paper discovers that there are safety-critical neurons, and these neurons might transfer to focus on reasoning after fine-tuning, which causes the safety misalignment.

**Strengths:**

(1) The studied problem is interesting, and the motivation is meaningful. It is important for LLMs to be safe. Understanding how LLMs’ safety misalignment emerges during fine-tuning could be very beneficial to future applications. and to ensure that LLMs still follow safety alignment after fine-tuning.

(2) This paper studies several related questions, providing a systematic understanding: It first identifies the RIM phenomenon in in-context CoT prompting and finetuning on reasoning tasks, then analyzes the mechanisms in in-context CoT prompting, and finally analyzes the mechanisms in finetuning.

(3) The paper chooses different appropriate methods/tools in different sections to investigate or analyze different perspectives.

(4) The experiments are conducted in various model families, providing convincing results. Experiment results support the discoveries.

**Weaknesses:**

(1) As mentioned in the strengths, this paper systematically analyzes reasoning-induced misalignment in different sections: identification of Reasoning-Induced Misalignment, RIM (Section 2); analyzing internal mechanisms of RIM in CoT in-context prompting (Section 3); and analyzing internal mechanisms of RIM in finetuning (Section 4). However, it lacks connections between the discoveries in different sections. For example, Section 2 discovers that effort-minimizing reasoning patterns could cause RIM, then a natural question is why effort-minimizing reasoning patterns cause RIM at the neuron level. But later sections didn’t mention this. Section 3 discovers that CoT inference changes the behavior of some attention heads, and then a natural question is whether fine-tuning changes the behavior of these attention heads in a similar way. But Section 4 didn’t study this. If my understanding is correct, Section 4 studies MLP neurons, but Section 3 studies attention heads. After reading Section 4, I am also wondering about these MLP neurons’ effects during CoT inference. Can we apply the methods used in Section 3 to also analyze fine-tuning mechanisms in Section 4? And can we apply the methods used in Section 4 to also analyze CoT inference in Section 3? It would be better if we could understand the neurons’ mechanisms in CoT inference and fine-tuning as a whole picture, and how these mechanisms affect effort-minimizing reasoning patterns.

(2) The experiments only test math reasoning ability. It would be more inspiring if we could also analyze other reasoning tasks.

**Questions:**

Please refer to the weaknesses.

---

> ### Author Response · Authors · 2025-11-20
>
> Thank you for your thorough comments and constructive suggestions. We also hope that our method could provide insights for the future research. Below are our responses to your concerns:
>
> ### __Q1: Connections between different sections.__
>
> We acknowledge the importance of maintaining a clear logic flow between sections, especially when we have multiple take-aways. We made the following clarification and enhancements.
>
> > __Q1.1: Why effort-minimizing reasoning patterns cause RIM at the neuron level?__
> >
> > In Section 4, we have the results in Figure 7 showing that effort-minimizing CoTs also lead to substantial neuron-level entanglement between safety and reasoning, i.e., they exhibit a larger RAS than the controlled CoT group.
>
>
> > __Q1.2: CoT inference changes after fine-tuning__
> >
> > We add the inference results of the fine-tuned models:
> > - The updated __Figure A5 & Figure A6__ show probe scores for refusal/fulfillment across different prompt tokens. Compared to Figure 3 (before FT), we observe a significantly smaller gap, i.e., an average relative decrease of  44.00% (0.34 before fine-tuning versus 0.19 post fine-tuning) after the 22nd layer between refusal and fulfillment plots, implying that FT weakens the representational capability of no-think tag in safety guardrails.
> > - __Figure A7__ shows the behavior of the identified refusal attention heads after FT.  We observe that these attention heads still exhibit a shift from the "instruction area" in think-mode to the “empty think” region (no-think). It implies that fine-tuning doesn’t affect the attention head much.
> > - In __Appendix C.3__- Attention Heads.  We further analyze the number of primary refusal attention heads (defined as the top 25% attention heads ranked by the percentage of samples exhibiting  refusal behavior): only a 0.8% drop after fine-tuning. It strengthens our conclusion that fine-tuning has limited impact on refusal-related  attention head behavior.
>
> > __Q1.3: Effects of ablation safety-related  MLP during inference__
> >
> > We have the results for causal intervention results on MLP neurons in Figure 6: Changes in misalignment rate (left) and math accuracy (right) when intervening on either the target (safety-related MLP neurons) or random MLP neurons. Results show that (i) intervention on the target neurons increases misalignment, and (ii) math reasoning accuracy is also highly associated with these safety-critical MLP neurons.
>
>
> ### __Q2: Results on other reasoning tasks.__
> > We have added a logic reasoning task (PrOntoQA) [1] and a commonsense reasoning task (StrategyQA) [2] to ensure the generalization of the RIM observation.
> >
> > From Table 1 (corresponds to __Table A8__ in the revised manuscript), we observe that fine-tuning on both logic and commonsense reasoning datasets generally leads to worse misalignment across the four models.
> >
> > &nbsp;
> >
> > __[Table 1]: Changes in misalignment rates after FT on four models.__
> | | Logic | Commonsense Reasoning |
> | ------ | ------ | ------ |
> | Qwen3-4B | +14.05% | +2.28% |
> | Phi3.5-Mini | +7.17% | +5.33% |
> | Mistral-7B | +7.13% | +8.90% |
> | OLMo2-1B | -5.10% | -4.06% |
> | __Average__ | __+5.81%__ | __+3.11%__ |
>
> >For the detailed RAS results after fine-tuning on multi-hop logical reasoning task, please refer to the response to Reviewer VT3w - Q2. In summary, RAS still demonstrates a strong correlation with catastrophic forgetting under the new logical reasoning task.
>
> Please let us know if you have further concerns. Thank you!
>
> References:
>
> [1] Language Models Are Greedy Reasoners: A Systematic Formal Analysis of Chain-of-Thought
>
> [2] Did Aristotle Use a Laptop? A Question Answering Benchmark with Implicit Reasoning Strategies

---

> ### Comment · Reviewer_mDjh · 2025-11-20
>
> Thank you for your efforts! I am sorry for having overlooked certain aspects in my initial review, and I appreciate the clarifications you have provided. I think the newly added experiments and explanations are valuable and strengthen the paper. They help construct a more complete and coherent understanding of the phenomenon. In particular, I now see that effort-minimizing reasoning patterns can lead to substantial neuron-level entanglement between safety and reasoning during fine-tuning. The identified attention heads and MLP layers contribute to Reasoning-Induced Misalignment (RIM) both during inference and fine-tuning. Moreover, the results from additional reasoning tasks further support the generalizability of the RIM phenomenon.
>
> Overall, I think this paper qualifies to be accepted as at least a poster at the conference, and I will keep my score at 8.
>
> That said, I remain hesitant to further increase my score. It will be better if there is more conceptual coherence. For example, it remains unclear how effort-minimizing reasoning patterns influence the activations of the identified attention heads and MLP neurons. It is also interesting if we could explore using the probing methodology used in Section 3 to analyze MLP neurons, or to use the methods in Section 4 to analyze attention heads, or if it is not very meaningful to do so, to explain why you chose the specific methods for attention heads and MLP neurons.

---

> ### Author Response · Authors · 2025-11-23
> **Response to follow-up question**
>
> Thank you for your thoughtful follow-up questions and suggestions to strengthen our paper's coherence.
>
> ### **Overall structure**
>
> As you outlined in the initial review, our paper follows this logic:
>
> In **Section 2**,  we report the observation of RIM in different scenarios:
> - During inference:
>   - think/no-think
>   - Effort-minising CoTs
> - Fine-tuning:
>   - Different reasoning tasks
>   - Effort-minising CoTs
>
> In **Section 3** and **Section 4**, we investigate the mechanisms underlying these observations during inference and fine-tuning, respectively.
> - We use attention in inference because attention weights can explain the importance of input tokens to the output, thus explaining why different input prompts lead to different safety behaviours without model parameter changes;
> - We study the changes of MLP as MLP is commonly related to new knowledge learning after fine-tuning[1,2].
>
> -----------
> More specifically,
>
> ### In **Section 3**:
> > we focus on the inference phrase, in which there are no parameter changes, only the input prompts differ. Therefore, our research question is
> >    - (i) which tokens in the input prompt are critical,
> >   - (ii) any internal mechanism causing it?
>
> To answer the two questions, we have:
> - [**Experiment part1**] Probe the refusal/fulfill behavior of different tokens in the think/no-think prompt. Note that the probes are not attention-head related, and are based on the complete hidden states of each token.
>    - [**Results**] Identify that the empty region between think tags is critical (Figure 3)
> - [**Experiment part2**] Identify attention heads that particularly focus on the "empty region" when generating the response.
>   - [**Why attention**, rather than other components such as MLP] It is commonly used to calculate the attention weights between input tokens and generated tokens, to analyze which input tokens contribute most to the outputs.
>   - [**Results in original version**] We identify so-called refusal attention heads (Figure 4), and our causal intervention experiment (Figure 5) shows they are indeed responsible for the refusal behavior.
>   - [**Results during rebuttal**] We have added probe results in Figures A5 & A6 and the attention head behavior in Figure A7 after fine-tuning.
> - [**ToDo**] The above experiments are specific to think/no-think modes, not effort-minimizing CoTs. Therefore, we are conducting experiments to:
>    - (i) identify critical tokens in effort-minimizing prompts that drive refusal behavior (analogous to Figure 3)
>    - (ii) localize attention heads that preferentially attend to these tokens (analogous to Figure 4)
>
> We will update the manuscript once these results are complete.
>
> ### **In Section 4** :
> > We focus on the changes in neurons after fine-tuning. We study MLPs as they are commonly associated with learned attributes/knowledge, see [1,2].
> - We show reasoning tasks increase entanglement, i.e., RAS in Figure 7.
> - We show effort-minimizing CoT increases entanglement, i.e., RAS in Figure 7 for both control and target groups, which considers both the MLP shrinkage in safety and growth in reasoning. Specifically, for Qwen3-4B, training with controlled CoT trajectories results in safety activation shrinkage (MLP activation) that is 27.66% smaller compared to training with target CoT trajectories.
>
> We hope this response clarifies the logic throughout the paper.  Please let us know if you are interested in further experimental results—any questions are very welcome.
>
> Thank you for your thoughtful engagement and the time you have dedicated to reviewing our work.
>
>
> References:
>
> [1] Transformer feed-forward layers are key-value memories
>
> [2] A Mechanistic Understanding of Alignment Algorithms: A Case Study on DPO and Toxicity

---

> ### Author Response · Authors · 2025-12-01
> **Follow-up experiments in updated PDF**
>
> We have updated the probe experiment for effort-minimizing CoT patterns in **C.4** (revised PDF). In general,
> - we have observed that the probe score gap (between reject and fulfill) over different input tokens are overall larger in control group, comparing that in target group. This implies that the input prompts in control group are better at triggering reject response when dealing with harmful inputs.
> - we have identified the important prompt tokens in control CoTs. For example, in the control group of the bias-driven shortcuts pattern, it highlights that we need to consider alternative assumptions for deliberative thought on safety issues.

---

### Author Response · Authors · 2025-12-01
**General response**

Dear reviewers and PCs/SACs/ACs:

We appreciate your time and effort in reviewing our submission, especially considering the recent administrative challenges.

Here is our **initial evaluation ratings**:
- Overall ratings (8 6 6), with confidence score as (4 3 4), all reviewers gave 3/3/3 to Soundness/Presentation/Contribution.

-----
Below is a summarisation of the rebuttal process:
### **Reviewer mDjh** (**Initial Rating**: 8).
*Comments and author responses*:
- Section Coherence:
   - We clarify the reasons of different research subjects in section 3 and section 4.
  - We add experiments of attention heads patterns for fine-tuning models.
  -  We add experiments to explain the efforts of “effort-minimising CoTs” during inference.
- Other reasoning datasets:
   -  We add experiments on two additional reasoning tasks, logic and commonsense reasoning.

**Reviewer Updates**: Overall, I think this paper qualifies to be accepted as at least a poster at the conference, and I will keep my score at 8.

---

### **Reviewer XKN8** (**Initial Rating**: 6).

*Comments and author responses*:
- I don't have major complaint about the paper.
  - We address their minor questions point-by-point, such as how many inputs for figure 3, new subsection title, and suggestion for gradient-norm based estimation.

**Reviewer Updates**: Thank you for the detailed response! I remain my original score.

---

### **Reviewer VT3w** (**Initial Rating** : 6) .

*Comments and author responses*:
- Other safety evaluation benchmarks
  - We add experiment results on two additional safety benchmarks, HarmBench and AgentHarm.
- Other reasoning datasets
  - We add experiments on two additional reasoning tasks, logic and commonsense reasoning.
- Other variants of the RAS metrics
  - We pointed out that we already have two variants, and add experimental results for additional four variants.

**Reviewer Updates**: I maintain my initial evaluation of this paper and lean to recommend its acceptance.

-------

Thanks, Authors.

---

### Meta-Review · Area_Chair_e6NN · 2026-01-06

**Summary:**

Focusing on the critical intersection of model safety and alignment with human values, the paper identifies a phenomenon termed Reasoning-Induced Misalignment (RIM), where enhanced reasoning capabilities inadvertently lead to misalignment. Through further analysis, the study uncovers the underlying causes of RIM by pinpointing specific attention heads and neurons that are simultaneously responsible for both reasoning and safety.

**Reviewer Concerns:**

Addressed Concerns:
- Generalization of Findings: The authors successfully addressed the concern raised by Reviewers mDjh and VT3w regarding the reliance on a single domain and safety benchmarkThe inclusion of additional reasoning tasks (PrOntoQA, StrategyQA) and safety benchmarks (HarmBench, AgentHarm) satisfied the reviewers that the phenomenon (RIM) is not limited to specific datasets.
- Confounders in Reasoning Patterns: Reviewer VT3w questioned whether linguistic features were confounding the "effort-minimizing" reasoning results. The authors performed a stratified analysis controlling for parse-tree depth, formal-marker density, and word counts, confirming that effort-minimizing patterns causally lead to greater misalignment regardless of these linguistic variables.

Outstanding Concerns：
- Deep Conceptual Coherence: Reviewer mDjh critiques that the paper lacked connections between inference (attention heads) and fine-tuning (MLP neurons) mechanisms, and still remains unclear after rebuttal.
- Metric Validation: The predictive utility of RAS versus standard methods or the depth of the mechanistic explanation might still be viewed as incremental.

**Reviewer Scores:**

- Reviewer mDjh (Original 8): Remains 8.
- Reviewer XKN8 (Original 6): Remains 6.
- Reviewer VT3w (Original 6): Remains 6.

---

### Decision · Program_Chairs · 2026-01-26

Accept (Poster)